# Protective Effects of *Salicornia europaea* on UVB-Induced Misoriented Cell Divisions in Skin Epithelium

**Natsumi Doi [1], Hiro Togari [1], Kenji Minagi [1], Koichi Nakaoji [2], Kazuhiko Hamada [2] and Masaaki Tatsuka [1,\*]**

[1] Department of Life Sciences, Faculty of Life and Environmental Sciences, Prefectural University of Hiroshima, Shoubara, Hiroshima 727-0023, Japan; q931003ca@ed.pu-hiroshima.ac.jp (N.D.); q923002fq@ed.pu-hiroshima.ac.jp (H.T.); q923005pa@ed.pu-hiroshima.ac.jp (K.M.)

[2] Research & Development Division, Pias Corporation, Kobe, Hyogo 651-2241, Japan; knakaoji@pias.co.jp (K.N.); khamada@pias.co.jp (K.H.)

[\*] Correspondence: tatsuka@pu-hiroshima.ac.jp; Tel.: +81-824-74-1756

**Abstract:** Correct orientation of cell division is extremely important in the maintenance, regeneration, and repair of continuously proliferating tissues, such as the epidermis. Regulation of the axis of division of epidermal cells prevents the apoptosis-induced compensatory proliferation, and eventually the cancer. Thus, the orientation of cell division is critical for maintaining the tissue architecture. In this study, we investigated the effects of *S. europaea* extract on the texture of human skin and the behavior of these cells during skin morphogenesis. In sun-exposed skin, *S. europaea* improved the texture. A multilayered, highly differentiated in vitro skin model indicated that, *S. europaea* extract suppressed the UVB-induced changes in the morphology of basal keratinocytes. Orientation of cell division was determined by measuring the axis of mitosis in the vertical sections of our experimental model. Analyses of the digital images revealed that *S. europaea* preserved the axis of division of basal keratinocytes from UVB-induced perturbations. Our findings uncover a new mechanism by which *S. europaea* responds to the spindle misorientation induced by UVB.

**Keywords:** *Salicornia europaea*; *Salicornia herbacea*; coral grass; UVB; sun exposure; skin care; epidermal structure; cell polarity; axis of division; mitosis

## 1. Introduction

Homeostasis and structural integrity of normal skin tissue are maintained by a balance between proliferation and terminal differentiation of keratinocytes. Once differentiated, these cells migrate across the stratum corneum and are eventually shed [1–3]. In stratified skin, symmetric cell divisions maintain the basal layer keratinocytes in a proliferative phase to preserve the pool of stem cells, while the asymmetric cell divisions along the apical-basal axis, supply the cells for keratinocyte differentiation [4–6]. Disorientation of these division axes create an imbalance between symmetric and asymmetric cell divisions, disturbing the tissue architecture of the skin, and its homeostasis [7,8] while, tight regulation of cell division axes inhibits abnormal cell growth and prevents apoptosis-induced compensatory proliferation [9]. Maintaining the normal orientation of cell division is critical for normal tissue homeostasis in the skin [10].

Exposure to UV radiation results in local skin damage [11,12]. The production of pigment by melanocytes partly protects the skin [13–15]. In the UV-irradiated keratinocytes, intracellular signals that regulate the cell fate in response to UV-induced DNA damage, are activated [16–19]. The pathways that control cell cycle checkpoint activation, DNA repair, and apoptosis are involved in skin homeostasis

and regeneration [20,21]. In addition, others, which are involved in maintaining structural homeostasis, are also modulated in response to UV and aging [22,23]. In particular, the cellular responses that disturb the epithelial integrity, are important as they lead to cosmetic problems, and even though considerable attention has been focused on them, the underlying mechanisms are not completely understood.

*Salicornia europaea*, also known as *S. herbacea*, is an edible halophytic annual dicot. In addition to culinary uses, it is also known to promote human health through its antioxidant, anti- inflammatory, antihyperglycemic, and antihyperlipidemic properties [24–26]. The aqueous extract of *S. europaea*, the so-called 'coral grass', is used in dermally applied cosmetics and is believed to improve the skin condition, but very few studies have addressed its effects on the epidermal keratinocytes. The mechanisms by which the extract helps maintain skin tissue homeostasis and its structural integrity remain largely unexplored.

To better understand how the aqueous extract of *S. europaea* protects the skin, we examined its effects on stress responses in the skin epidermis, using both in vivo and in vitro assays. We found that *S. europaea* improved the texture of sun-exposed human skin significantly. In a multilayered, highly differentiated in vitro skin model, *S. europaea* could prevent the UVB-induced aberrant stratification by blocking the distortions in the axis of division of the basal keratinocytes. The prevention of the misorientation of the axis of division is suggested as a novel mechanism of *S. europaea* in improving the skin texture.

## 2. Materials and Methods

### 2.1. S. europaea Extract

The aqueous extract of *S. europaea*, the so-called 'coral grass' that is extracted from the sangoso plant in Akkeshi-cho, Hokkaido, Japan, was purchased from Presperse Corporation (Somerset, NJ, USA). This material, which was extracted using 30% 3-methoxy-3-methyl-1-butanol, is supplied as the raw ingredient for cosmetics, with a robust quality assurance providing the manufacturing quality and traceability data sheet (https://www.knowde.com/products/presperse-coral-grass), and is commercially available.

### 2.2. Subjects

The study involved 29 healthy Japanese females, aged between 30 and 55 years, and without acute or chronic diseases, including skin diseases. All of them voluntarily signed the informed consent form, and were available for follow-up during the testing period. This research, performed on humans, was complied with the principles of the Declaration of Helsinki and Japan, and was reviewed and approved by the Institutional Review Board in the legally incorporated medical institution 'Kenshokai', Osaka, Japan (Permission number: 2017-9) based on Japanese Guidance for the Safety Evaluation of Cosmetics 2015.

### 2.3. S. europaea Treatment

Cream (ingredients: purified water, mineral oil, petrolatum, cetostearyl alcohol, propylene glycol, sodium lauryl sulfate, isopropyl palmitate, imidazolidinyl urea, methylparaben, and propylparaben), containing *S. europaea* extract (3% solution), was topically applied on the skin of the study subject, in the sun-exposed area, on one side of the face, and simultaneously placebo controlled trial was carried out with the same individual, on the contralateral side of the face (Figure 1A). All volunteers underwent twice-daily treatments for 4 and 8 weeks from September to November, and the test areas were exposed to sun for 3 h a day (7:00–8:30 a.m., and 12:30–14:00 p.m.). During the test period, none of the volunteers used any skin formulation, containing steroids or any compounds, other than the test agent on the skin. No adverse events, including itching, erythema, or others, at the test area, were observed in any of the volunteers, and none of them had any skin problems that hindered the evaluation.

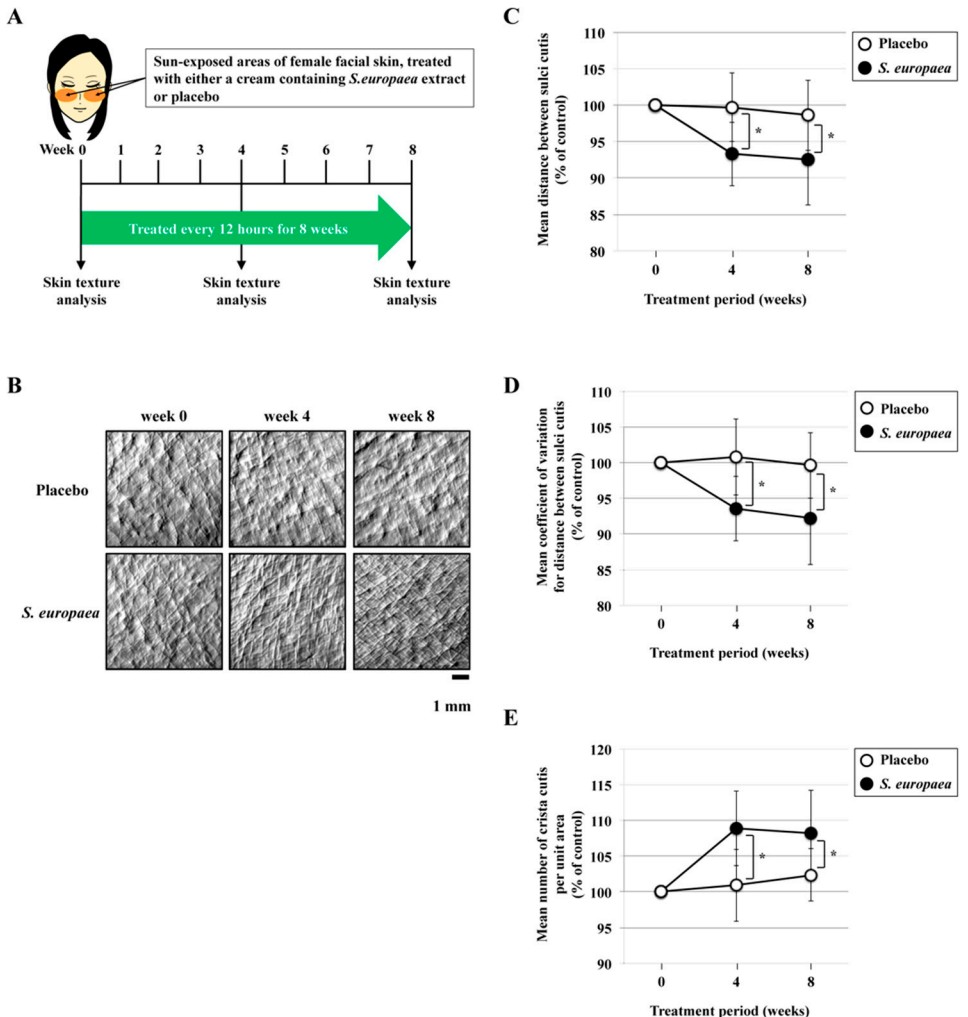

**Figure 1.** Improvements in the texture of the sun-exposed skin after treatment with *S. europae*. (**A**) Study design and treatment schedule. A cream containing *S. europaea* extract was topically applied to the facial skin in sun-exposed areas of 29 healthy volunteers, and simultaneously a placebo-controlled trial was carried out with the same subjects, on the contralateral side of the face. All volunteers underwent twice-daily treatments for 4 and 8 weeks. The skin texture was analyzed before and at 4 and 8 weeks after the treatment. (**B**) Typical reflective replica images from skin treated either with *S. europaea* extract, or placebo. The images were obtained using 3D Skin Roughness Analysis Measurement System. Scale bar, 1000 μm. (**C**) Effects of *S. europaea* on the distance between sulci cutis. The data were obtained from three different areas (50 mm × 50 mm) of each skin sample from 29 subjects. Values indicate means ± S.D. * $p < 0.02$. (**D**) Effects of *S. europaea* on coefficient of variation between sulci cutis in the above data. Values indicate means ± S.D. * $p < 0.007$. (**E**) Effects of *S. europaea* on number of crista cutis per unit area. The data were obtained from three different areas (50 mm × 50 mm) of each skin sample from 29 subjects. Values indicate means ± S.D. * $p < 0.005$.

## 2.4. Skin Texture

A reflective replica analysis system for skin surface, the 3D Skin Roughness Analysis Measurement System (ASA-03RXD, Asahi Biomed Co., Kanagawa, Japan), was used to study the effects of *S. europaea* treatment. The system allows the quantitative evaluation of skin surface texture in a non-invasive manner [27,28]. According to the manufacturer's instructions, skin texture was evaluated using three parameters: (i) distance between the sulci cutis, (ii) the coefficient of variation between the sulci cutis, and (iii) number of crista cutis per unit area. The decreased distance, decreased coefficient of variation

for distance, and increased number of crista cutis indicated improved skin. The data were obtained from three different areas (50 mm × 50 mm) of each skin sample.

## 2.5. Analyses with a Multilayered, Highly Differentiated In Vitro Skin Model

We tested various commercially available, in vitro human skin models and found EPI-200 (MatTek, Ashland, MA, USA) to be the most suitable and the most reliable model for multilayered, highly differentiated skin. The 3-D cultured cells were exposed to UVB (100 mJ/cm$^2$ and 200 mJ/cm$^2$), followed by treatment with *S. europaea* extract (0.03%, 0.1%, and 0.3%). The controls maintained included sham-exposed and sham-treated cultures. The medium was changed every 2 days with or without *S. europaea* extract. The 3-D cultured cells were harvested after 10 days, formalin-fixed, paraffin-embedded, and serially sectioned. The cross-sections were stained with hematoxylin and eosin and studied under a light microscope. Three independent experiments were performed, and the axis of division was measured in more than 25 cells in metaphase in each experiment, for a total of 80 observations.

## 2.6. Statistics

The paired Student's t-test was used to detect significant differences between *S. europaea*-treated test groups and untreated control groups (Figure 1). Significance of variance of the estimated directions in Figure 3B was analyzed using the F-test.

## 3. Results

### 3.1. Effects of S. europaea on Skin Texture in Sun-Exposed Areas

Aqueous extract of *S. europaea* was topically applied to the sun-exposed areas of the facial skin (Figure 1A) to evaluate its protective effects. After 4 and 8 weeks of twice-daily treatments, the results were quantitatively assessed using a reflective replica analysis system, designed to detect the three-dimensional skin roughness. When treated with *S. europaea,* the skin surface texture significantly improved, compared to the placebo group (Figure 1B), resulting in a densely packed furrows of sulci cutis (Figure 1C). We also found that the variation in the distance between the sulci cutis decreased, smoothening the rough and uneven texture of the skin (Figure 1D). In addition, the number of crista cutis increased in the treated skin (Figure 1E). Thus, *S. europaea* improved the textural characteristics of the sun-exposed skin significantly.

### 3.2. Effects of S. europaea on the Morphology and Arrangement of Epidermal Cells Located in the Basal Layer Exposed to UVB

Analyses of the morphology and arrangement of epidermal cells in the basal layer highlight the involvement of basal cells in maintaining the normal tissue homeostasis and structural integrity of the skin. To study the effects of *S. europaea* on UVB-induced morphological alterations in basal keratinocytes, we used a multilayered, highly differentiated in vitro skin model. Morphological analyses of keratinocytes in this system revealed that UVB radiation distorted the cuboidal morphology of basal keratinocytes, and the extract of *S. europaea* largely suppressed these morphological changes (Figure 2).

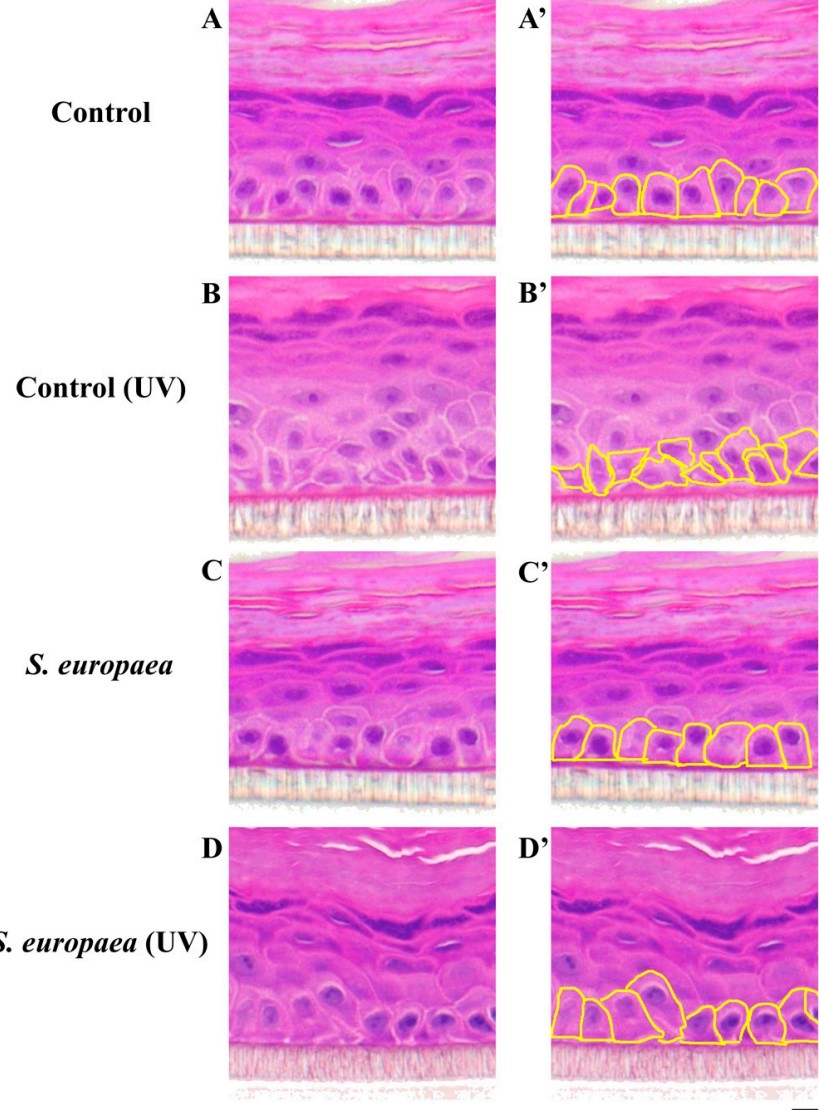

**Figure 2.** Improvements in the morphology and arrangement of UVB-exposed epidermal cells in the basal layer, after treatment with *S. europaea*. Light microscopic images of the cross section of hematoxylin-eosin-stained keratinocytes in a multilayered, highly differentiated in vitro skin model. (**A**) control keratinocytes; (**B**) UVB-exposed keratinocytes; (**C**) control keratinocytes after treatment with *S. europaea*; and (**D**) UVB-exposed keratinocytes after treatment with *S. europaea*. Typical morphological features of suprabasal keratinocytes are demarcated with yellow lines (**A'**–**D'**). Scale bar, 10 μm.

### 3.3. Effects of S. europaea on the Oriented Cell Division of Basal Keratinocytes Post UVB Exposure

Based on these observations, we speculated that *S. europaea* extract preserves the normal balance between the symmetric divisions (parallel to the basement membrane, to form stem cells) and asymmetric divisions (perpendicular to the basement membrane, to produce the differentiated cells) of epidermal cells in the basal layer, which were disturbed by UVB-radiation. The orientation of the axis of division of mitotic basal keratinocytes was quantitatively analyzed in a multilayered, in vitro skin model, as shown in Figure 3A. The distribution of angles of division of basal keratinocytes is presented in a scatter plot (Figure 3B) and in a Rose plot (Figure 3C). Statistical analysis revealed that UVB radiation disturbed the normal, homeostatically regulated balance between the symmetric and asymmetric cell divisions, leading to a decrease in the variance of the division angle distributions, and that *S. europaea* could suppress such disruptions.

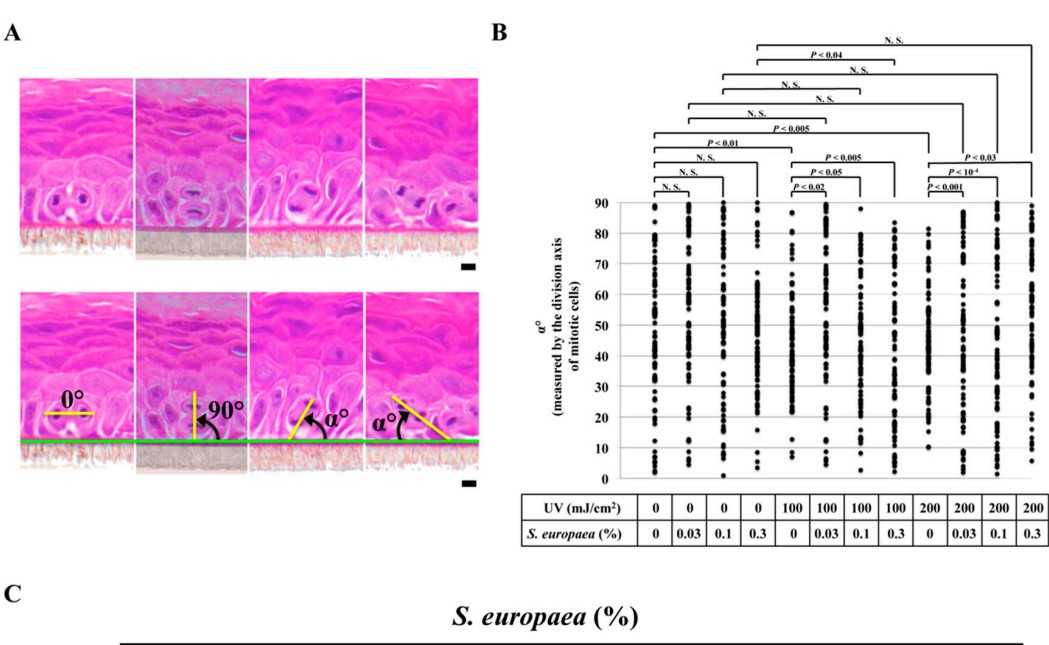

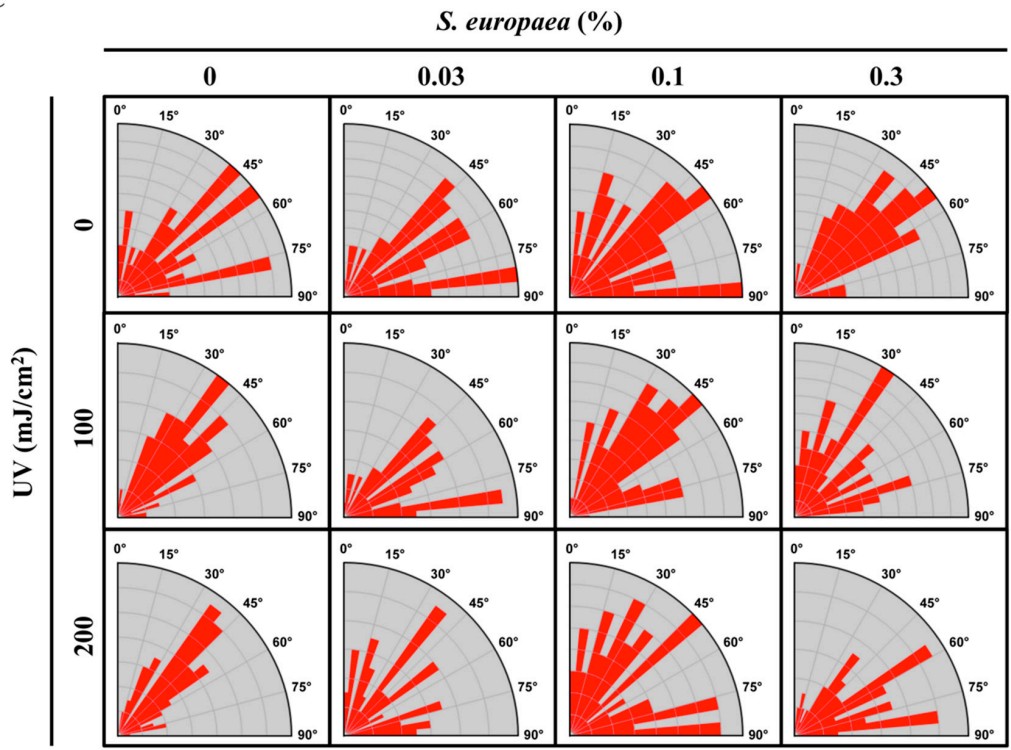

**Figure 3.** Improvements in the directional cell division of UVB-exposed epidermal cells located at the basal layer, after treatment with *S. europaea.* (**A**) Angle definition of cell divisions in suprabasal keratinocytes. Light microscopic images of the cross section of hematoxylin-eosin-stained mitotic suprabasal keratinocytes in a multilayered, highly differentiated in vitro skin model are shown in the upper panels. Here, we defined the angles for quantitatively characterizing the orientation of the axis of cell division in mitotic basal keratinocytes as shown in the lower panels. Scale bar, 10 μm. (**B**) Dot plot of the division angle ($\alpha°$) of suprabasal keratinocytes. Each symbol in the dot plot represents the angle measured from three different experiments each, observed in over twenty-five metaphase cells (total 80 observations). UVB-irradiated or non-irradiated keratinocytes were treated with different concentrations of S. europaea, as indicated. The variance of $\alpha°$ was found to be not significantly different between unexposed, untreated cells and unexposed, *S. europaea*-treated cells. On the other hand, the variance of $\alpha°$ was significantly different among the unexposed cells and UVB-exposed cells and among the unexposed and untreated cells and UVB-exposed, *S. europaea*-treated cells. (**C**) Rose plots for (**B**).

*3.4. Effects of S. europaea on Stratification during Skin Differentiation after UVB Eexposure*

We examined the effects of *S. europaea* on the stratification of UVB-exposed skin, during differentiation. Sheet-like layered structures in our in vitro skin model became morphologically disturbed and uneven upon UVB exposure, and these disruptive changes were prevented by *S. europaea* (Figure 4).

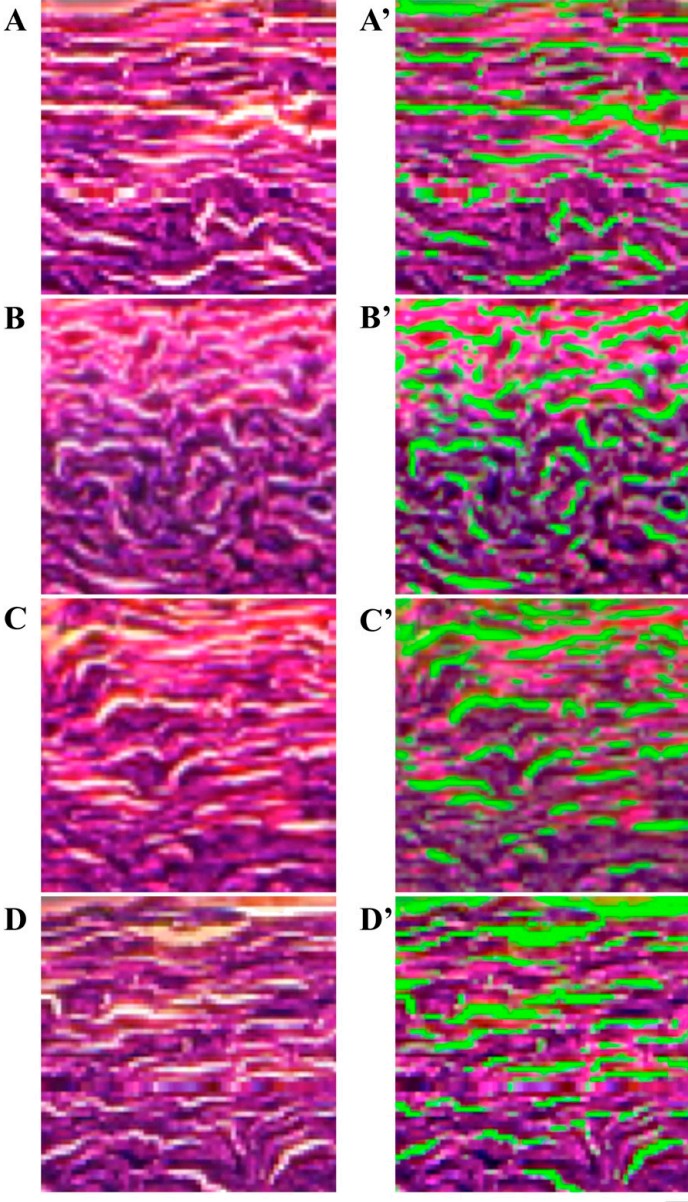

**Figure 4.** Improvements in stratification of UVB-exposed skin during differentiation, after treatment with *S. europaea* extract. Light microscopic images of the cross section of hematoxylin-eosin-stained keratinocytes located in the stratified region (the upper region) of a multilayered, highly differentiated in vitro skin model. (**A**) Control keratinocytes; (**B**) UVB-exposed keratinocytes; (**C**) control keratinocytes after treatment with *S. europaea*; and (**D**) UVB-exposed keratinocytes after treatment with *S. europaea*. Typical morphological features in the stratification are indicated with green lines (**A′–D′**). Scale bar, 10 µm.

## 4. Discussion

The epidermal structure, important for many functions, is damaged by exposure to UV. Efficient control of photodamage, and prevention of the ensuing premature skin aging, are some of the major issues in the development of cosmetics. Active ingredients, with UV absorbing properties to reduce the amount of radiation penetrating the skin, and those with antioxidant, anti-inflammatory, and immunomodulatory effects, are used in cosmetics and personal care products [29–32]. Ingredients that can reduce skin pigmentation, inhibit melanin biosynthesis and block the melanin-uptake by keratinocytes, are important in skin lightening and in treatment of hyperpigmentation [33–35]. In this context, skin problems such as impaired epidermal barrier function, reduced skin elasticity, abnormal keratinization, and rough skin texture, which are caused by UV in the sun-exposed areas, need to be addressed effectively while formulating cosmetics, in addition to the above-mentioned features [36].

To maintain its integrity and function, the healthy skin undergoes continual rejuvenation through the steady supply of terminally differentiated keratinocytes from the basal layer. Proliferation of skin stem cells in the basal layer is critical for this process [37]. Asymmetric cell divisions, in which the axis of mitotic spindle orientation is perpendicular to the basement membrane, supply the stratified cells to be terminally differentiated, while the symmetric cell divisions, where the mitotic spindles are oriented parallel to the basement membrane, produce stem cells to replenish in the basal layer [6,38]. Recently, several lines of evidence have indicated that the misorientation of the axis of cell division causes defects in the stratification and differentiation of keratinocytes in the skin epithelium [7,8,38–40]. In agreement with this concept, we observed that UVB disrupts stratification in the in vitro skin model. While mechanisms of how the genotoxic stress response pathways connect to the axes of division remain undefined, it is likely that genotoxic stress response pathways control the symmetric and asymmetric cell divisions in skin stem cells. UVA radiation also may induce DNA damages, and possibly elicit a similar response.

In this work, we demonstrated that treatment with *S. europaea* extract helps improve the skin texture. Our in vitro skin model showed the importance of the direction of mitotic spindle orientation in the stem cells in maintaining skin homeostasis. Prevention of the misorientation of the axis of division in skin stem cells should be a target while developing novel active ingredients for cosmetics. To our knowledge, the action of *S. europaea*, in skin smoothening and healing, is the first example of such a mechanism to promote the fidelity of mitotic direction.

*S. europaea* extract is likely to have ingredient/s which can inhibit the UVB-induced disruption of oriented cell divisions. The establishment of cell polarity is critical for the form and function of various epithelial lineages to support the tissues. Recent advances in understanding the regulation of cell polarity have shown the biochemical and molecular basis of cell polarization and migration [41,42]. Among the known multilevel regulators, Cdc42, one of the Rho family GTPases, is critical for cell polarity, and loss of its activity leads to disoriented cell divisions [42]. Its effects on skin differentiation in presence of UVB, are currently unknown. *S. europaea* extract contains water-soluble factors that are probably needed in halophytic regulation, involving complex salt-inducible signaling related to small GTPase regulation [43]. Further studies are required to determine if UVB inactivates the Cdc42, and whether *S. europaea* extract can restore it.

In the previous studies, *S. europaea* is shown to have diverse biological activities, including antioxidant [25,26,44], anti-inflammatory [25,45], hypolipidemic and anti-hyperglycemic [24,46], anti-hypertensive [47], activities and can regulate osteoblast and adipocyte differentiation [48], and cancer invasion [49]. Possible active ingredients, including several flavonoid glucosides and flavonoid glycosides, have been identified in *S. europaea* [50,51], and these may, at least partly, account for its diverse biological activities, while other possible bioactive components are yet to be explored. The key ingredients of *S. europaea* extracts responsible for promoting the skin texture, are also not known but are certainly promising candidates for improving the skin tone, probably with novel mechanisms of action. Indeed, recent studies support the notion that *S. europaea* extracts contain key ingredients for the skin protective activities, among various halophyte plant extracts [52–54].

## 5. Conclusions

This work shows that a water-soluble extract of *S. europaea* has the potential to improve the texture of the skin in the sun-exposed areas, by suppressing the misoriented axes of cell division in the proliferative basal layer of the skin, suggesting a novel approach to skin care. Little is known about the UV-induced perturbations of axis of division, and how this affects the skin texture. Further studies are needed to understand the mechanisms underlying the positive effects of *S. europaea*.

**Author Contributions:** K.N., K.H., and M.T. designed the experiments; N.D., H.T., K.M., and M.T. performed the experiments; N.D., K.N., and M.T. analyzed the data; N.D., K.N., K.H., and M.T. wrote the manuscript. All authors have read and agreed with the version of the manuscript.

**Funding:** This study was supported by JSPS KAKENHI Grant Number JP17K00556, and grants from the Prefectural University of Hiroshima (Interdisciplinary and Priority Research (S) H-30 and H-31) and the Prefectural University of Hiroshima Graduate School of Comprehensive Scientific Research to Natsumi Doi (Research Associate Grant H-30 and H-31).

**Acknowledgments:** We thank Takahide Ota for critical comments on the work and also thank Yuuki Kunimatsu and Kouhei Fujiura for their technical assistance.

**Conflicts of Interest:** The authors declare no conflicts of interest.

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
