# Peer review of "Protective Effects of Salicornia europaea on UVB-Induced Misoriented Cell Divisions in Skin Epithelium"

_cosmetics, doi:10.3390/cosmetics7020044_

Round 1

Reviewer 1 Report

The manuscript "Protective effects of Salicornia europaea on UVB-induced misoriented cell divisions in skin epithelium" is well written and presented nicely.

I do not have any specific suggestions for the authors.

Author Response

Thank you for your e-mail of 2-Jun-2020, regarding our manuscript (Ref.: cosmetics-790453 ‘Protective effects of Salicornia europaea on UVB-induced misoriented cell divisions in skin epithelium’ by Natsumi Doi et al.), together with the comments from the Reviewers. We were pleased to learn that the Cosmetics is interested in publishing our manuscript upon revision. The comments made by the reviewers were most helpful and gave us a better perspective of our work. The following revisions have been made:

1. Reviewers #2: As suggested by Reviewers #2, more recent references regarding protective effects by and active ingredients of Salicornia europaea have been added in the lines 186-194.

2. Reviewers #3: As suggested by Reviewers #3, we have improved the 'Materials and Methods' section as follows: i) More detail information about the Salicornia europaea extract has been added. ii) The details of the Institutional Review Board has been described. iii) More detailed information about skin texture analysis has been provided. iv) The 'Materials and Methods' section has been put before results. v) Figure 5 has been removed.

We would like to thank the Reviewers for their helpful comments and hope that the revised manuscript is suitable for publication in the 'Cosmetics'.

Yours sincerely,

Masaaki Tatsuka

Reviewer 2 Report

The article is well organized scientifically sound and reviews the current techniques being used in meausuring the protective effects of the aqueous extract o Salicornia europae on UVB-induced misoriented cell divisions in skin epithelium. 

I would recommend to the authors to extend the paragraph between the lines 186-194, with more recent references on the skin protective activities and key ingredients of S. europae.

Author Response

(The authors gave the same response as above.)

Reviewer 3 Report

  Thank you very much for submitting your Manuscript cosmetics-790453 entitled " Protective effects of Salicornia europaea on UVB-induced misoriented cell divisions in skin epithelium" for Cosmetics.  I appreciate your excellent data on protective effect of a plant extract on UVB-induced skin epithelium. I welcome the beautiful figures. According to my suggestions, I wish the authors will revise them soon.

  • M & M is preferred to put before results.
  • An effect of plant extract is affected by regions and terms of a plant and that of an extract causes large variation by the extract way and solvents.   So, the analytical information of the extract method is important. I wish the authors make a clear the analytical information of your extract in results.  
  • Hopefully, a certification number of volunteer study approved by the Institutional Review Board should be described at 4.2.
  • More detailed information is provided with skin texture at 4.4. Based on this information, it is difficult to check reproducibility of the volunteer study.
  • Unfortunately, the information of fig.5 is poor in results and be not touched in discussion. I think this hypothesis is unreasonable based on the limited data in this manuscript. I recommend you to delete this figure.

Thank you.

Author Response

(The authors gave the same response as above.)
